# Study protocol: establishment of a multicentre pre-eclampsia database and biobank in Sweden: GO PROVE and UP MOST, a prospective cohort study

Lilja Thorgeirsdottir,[1] Malin Andersson,[2] Ove Karlsson [ORCID],[2] Sven-Egron Thörn,[2] Jonatan Oras,[2] Verena Sengpiel [ORCID],[3,4] Teresia Svanvik [ORCID],[3,4] Helen Elden [ORCID],[1,4] Karolina Linden [ORCID],[1] Katja Junus [ORCID],[5] Susanne Lager [ORCID],[5] Ida Enskär,[5] Teelkien van Veen,[6] Johan Wikström [ORCID],[7] Isabella Björkman-Burtscher,[8] Anna Stigsdotter Neely,[9,10] Anna-Karin Wikström,[5] Lina Bergman [ORCID] [3,4,11]

For numbered affiliations see end of article.

**Correspondence to**
Dr Lina Bergman;
lina.bergman.2@gu.se

## ABSTRACT

**Introduction** Pre-eclampsia, a multisystem disorder in pregnancy, is one of the most common causes of maternal morbidity and mortality worldwide. However, we lack methods for objective assessment of organ function in pre-eclampsia and predictors of organ impairment during and after pre-eclampsia. The women's and their partners' experiences of pre-eclampsia have not been studied in detail. To phenotype different subtypes of the disorder is of importance for prediction, prevention, surveillance, treatment and follow-up of pre-eclampsia.

The aim of this study is to set up a multicentre database and biobank for pre-eclampsia in order to contribute to a safer and more individualised treatment and care.

**Methods and analysis** This is a multicentre cohort study. Prospectively recruited pregnant women ≥18 years, diagnosed with pre-eclampsia presenting at Sahlgrenska University Hospital, Uppsala University Hospital and at Södra Älvsborgs Hospital, Sweden, as well as normotensive controls are eligible for participation. At inclusion and at 1-year follow-up, the participants donate biosamples that are stored in a biobank and they are also asked to participate in various organ-specific evaluations. In addition, questionnaires and interviews regarding the women's and partner's experiences are distributed at follow-up.

**Ethics and dissemination** By creating a database and biobank, we will provide the means to explore the disorder in a broader sense and allow clinical and laboratory discoveries that can be translated to clinical trials aiming at improved care of women with pre-eclampsia. Further, to evaluate experiences and the psychological impact of being affected by pre-eclampsia can improve the care of pregnant women and their partners. In case of incidental pathological findings during examinations performed, they will be handled in accordance with clinical routine. Data are stored in a secure online database. Biobank samples are identified through the women's personal identification number and pseudonymised after identification in the biobank before analysis.

This study was approved by the regional ethical review board in Gothenburg on 28 December 2018 (approval number 955-18) and by the Swedish Ethical Review Authority on 27 February 2019 (approval number 2019-00309).

Results from the study will be published in international peer-reviewed journals.

**Trial registration number** ISRCTN13060768

## Strengths and limitations of this study

► This study is a national three-centre study, enabling inclusion of women from different socioeconomic backgrounds and by the size of the units, permits a large biobank and database of pre-eclampsia in a short timeframe.

► The interdisciplinary team of obstetricians, midwives, anaesthesiologists, radiologists and cardiologists and collaboration between units from developed and developing countries contribute to a wider understanding of the disease from different aspects.

► The study uses predefined variables by an international consensus of predictors and outcomes in pre-eclampsia research—enabling merging of data with other international research groups.

► The hospital integrated biobanks ensure a high quality and standardised methods for the biological samples included in the study.

► Women with pre-eclampsia are delivered (as a cure) often before onset of complications and thus the endpoints will often consist of 'preterm delivery' instead of true maternal organ complications.

## INTRODUCTION

Pre-eclampsia is one of the most serious complications of pregnancy affecting 3%–8% of pregnancies worldwide.[1 2] Pre-eclampsia leads to more than 60 000 maternal deaths annually and is also a common cause of maternal and neonatal morbidity.[2 3] Pre-eclampsia is a multiorgan disorder involving

the brain, cardiovascular system, fetoplacental unit, haematological system, kidneys, liver, lungs and maternal vessels.[1] It may cause heart failure, pulmonary oedema, renal failure, haemostatic disorders, cerebrovascular events, seizures (eclampsia), intrauterine growth restriction and intrauterine death. The pathophysiology of pre-eclampsia is not completely understood, but the hypothesis is based on placental dysfunction, in particular in early onset disease (diagnosis <34 gestational weeks). In late onset disease, maternal factors such as diabetes, cardiac disease, multifetal pregnancy and obesity are thought to play a more important role.[4] There is a paucity of data on how to evaluate threatening organ dysfunction in pre-eclampsia. At present, there are only a few tests available mainly regarding liver, coagulation and renal function.[5] Clinicians lack predictors or objective assessment of cardiovascular, endothelial or neurological dysfunction in pre-eclampsia. An increased understanding of the underlying pathophysiology of organ complications and increased accuracy regarding prediction would allow individualised treatment. This includes magnesium sulfate for prevention of seizures, targeted blood pressure treatment for prevention of intracranial haemorrhage, customised administration of regional anaesthesia, particularly in cases with disturbed coagulation and targeted treatment in case of cardiac dysfunction. By identifying biomarkers with high sensitivity and specificity for various threatening organ complications, end-organ complications could be identified and managed in time and some cases of iatrogenic preterm labour could be avoided, potentially improving the short-term and long-term outcomes for babies born to women with early onset pre-eclampsia.[6]

The capacity to recognise, exploit and phenotype different subtypes of pre-eclampsia is of importance for prediction, prevention, surveillance, treatment and follow-up of pre-eclampsia. This protocol adheres to the international consensus about standardised data and biological sample collections in pre-eclampsia research.[7] Adherence to such standardised protocols will facilitate the cooperation with other research teams and hopefully improve the understanding of the causes of pre-eclampsia and development of targeted treatment strategies.

The women's and their partners' experiences of pre-eclampsia during pregnancy and post partum has not been studied in detail. Qualitative interview studies with women with pre-eclampsia have described pre-eclampsia as a fearful, powerless, painful and/or life-threatening event. They have also described inadequate clinical management and care and many women felt lucky to have survived.[8 9] By studying how women and their partners experience pre-eclampsia and the consequences regarding health-related quality of life, self-efficacy, anxiety, childbirth experiences and breast feeding, we aimed to improve personalised care and follow-up after a pregnancy complicated by pre-eclampsia.

## METHODS/DESIGN

This is a Swedish multicentre prospective cohort study prospectively including women diagnosed with pre-eclampsia and their partners and in addition normotensive controls. A flowchart of the study is presented in figure 1. The study started to include women during the fall of 2019 and will continue to recruit until 2023.

### Aims

The overall aim of this study is to establish a database and biobank including two Swedish University pre-eclampsia cohorts: GO PROVE (Gothenburg Preeclampsia Obstetric adverse events) and UP MOST (Uppsala Pregnancy Complication Study) for pre-eclampsia research, to facilitate laboratory-based, translational and clinical research on pre-eclampsia and to contribute to a safer and more patient-centred care of women with pre-eclampsia in the acute and long-term perspective, focused on specific organ involvement. Thus, the following actions are included in the project:

► To establish a biobank and database with core predictors and outcomes for pre-eclampsia research in Sweden (GO PROVE and UP MOST).
► To examine alterations in cerebral structure and function through magnetic resonance (MR) investigation of the brain, Doppler examination of cerebral blood flow, cerebral biomarkers, electroencephalograms (EEG) and neurocognitive assessment of mental processing speed, working memory, executive functions and episodic memory in women with pre-eclampsia compared with normotensive controls at diagnosis and at 1 year post partum (GO PROVE and UP MOST).
► To examine cardiac impairment and prediction of cardiac impairment through echocardiography and cardiac biomarkers in women with pre-eclampsia compared with normotensive controls at diagnosis and at 1 year post partum (GO PROVE).
► To examine vascular dysfunction through measurement of peripheral resistance and blood-based endothelial biomarkers in women with pre-eclampsia compared with normotensive controls at diagnosis and at 1 year post partum (GO PROVE).
► To examine coagulation profiles through blood-based biomarkers and tests such as thromboelastometry (ROTEM) and impedance aggregometry (Multiplate) in women with pre-eclampsia compared with normotensive controls (GO PROVE)
► To increase the knowledge and understanding of women's and their partners' experiences of pre-eclampsia during pregnancy, birth and post partum through validated measurements and qualitative interviews (GO PROVE).

### Population

Pregnant women ≥18 years of age with a diagnosis of pre-eclampsia after gestational week 20, presenting at Sahlgrenska University Hospital (10 500 deliveries/year),

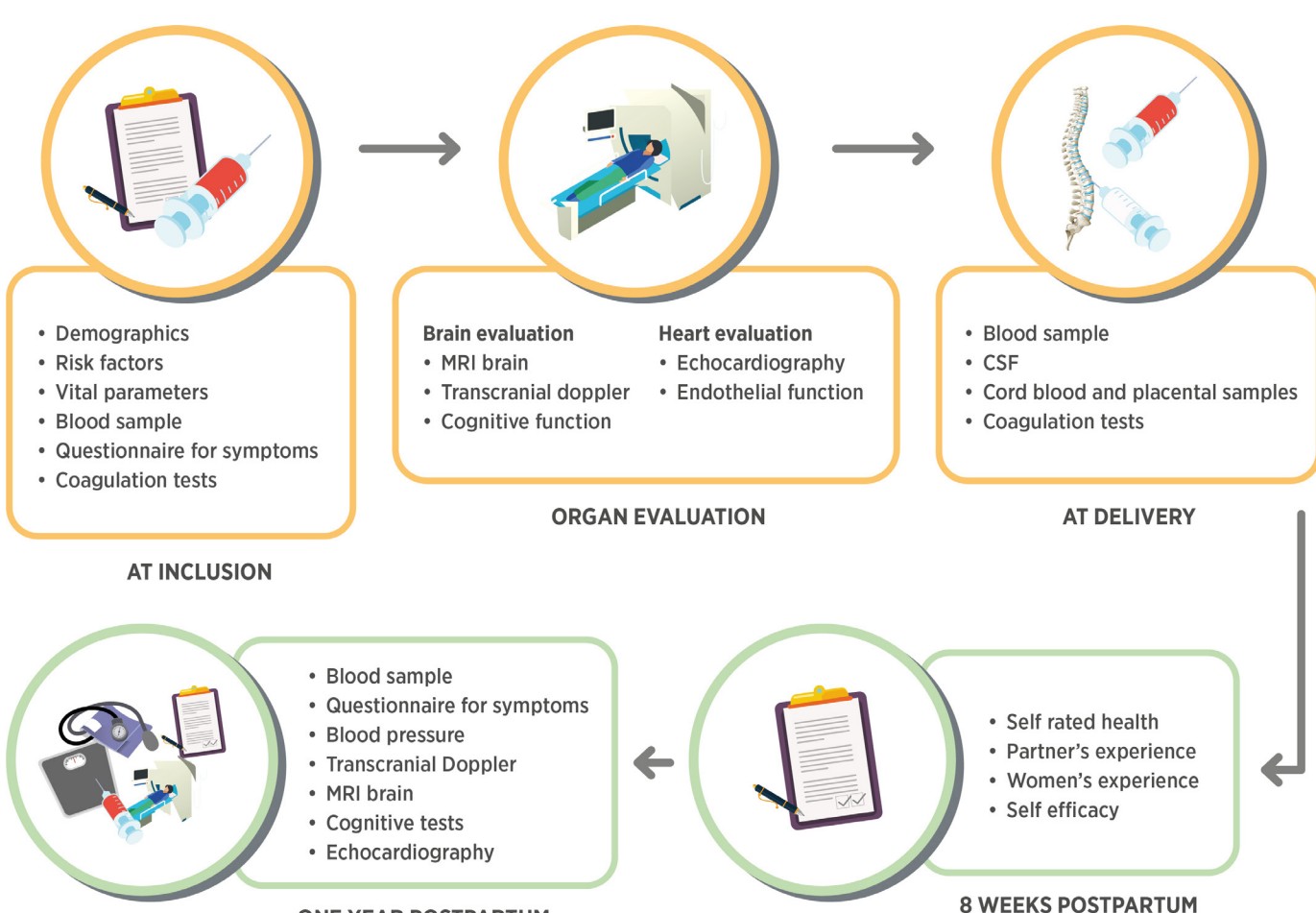

**Figure 1** Flowchart of the study. Investigations and biological samples from inclusion to 1 year post partum. CSF, cerebrospinal fluid.

Uppsala University Hospital (4200 deliveries/year) and Södra Älvsborgs Hospital (3100 deliveries/year) are approached for inclusion. Both participating women and their partners are eligible for the qualitative interview study. Healthy women ≥18 years with normotensive pregnancies are included as controls.

### Exclusion criteria

Women unable to understand oral and written information and unable to give written informed consent are not eligible as well as women with pre-existing hypertension, diabetes mellitus before or during pregnancy and pre-existing cardiovascular, renal or cerebral disease. MR contraindications for MR group includes claustrophobia. Women who have previously been included in the study will not be eligible for their next pregnancy.

For the database and biobank as a whole, there will be no matching. For specific substudies, there might be matching on a group level for gestational age at examination.

The diagnosis of pre-eclampsia will be established according to Swedish guidelines for hypertensive disorders in pregnancy (www.sfog.se) that generally follows international guidelines such as the international society for the study of hypertension in pregnancy (ISSHP), the

National Institute for Clinical Excellence (NICE) and the American College of Obstetricians and Gynecologists (ACOG) with the addition that we will require proteinuria to be present to limit the heterogeneity of the population.

### GO PROVE and UP MOST database

The database is provided by MedSciNet (MedSciNet AB, Stockholm, Sweden) and includes exposure and outcome variables as proposed for pre-eclampsia research by the international collaborative group for pre-eclampsia research Co-Lab.[7] In addition, specific variables for this study have been included (online supplemental table S1 and box 1). Biochemical tests when performed in routine healthcare are also collected, in addition to biochemical tests analysed specifically for the study, and are presented in table 1.

### GO PROVE and UP MOST biobank

GO PROVE and UP MOST use the Hospital Integrated Biobank (SIB, www.biobanksverige.se) that enables samples to be handled in a standardised manner through the hospitals' clinical laboratories, thus ensuring sample quality. Samples are sent to the local hospital's clinical chemistry laboratory or directly to the hospital biobank facility, where they are spun, aliquoted into 225 µL wells

**Box 1 Neurological symptoms at inclusion, at MRI brain and at follow-up 1 year post partum**

Oedema (n)
Visual disturbances (b)
*No visual disturbances* (b)
*Blurred vision* (b)
*Double vision* (b)
*Temporary loss of vision* (b)
*Sensitivity to light* (b)
*Partial loss of vision or blind spots in normal field of vision* (b)
*Flickering lights* (b)
*Other* (n)

**Time of onset for visual disturbances (n)**
*N/A (no visual disturbance)* (b)
Headache (o)
Headache onset (o)
Epigastric/abdominal pain (o)
Tightness in chest (b)
Shortness of breath (b)
Focal neurological deficits (n)
Tendon reflexes (n)
Nausea (b)
Vomit (b)
Form of confusion (b)
Twitching or jerking in arms and/or legs (b)
Has the woman's family/friends experienced the patient as not responding or staring blankly into space (b)
Difficulty concentrating (b)
Speech affected (b)
Hearing affected (b)
Mood changes (n)
Other mood changes than listed (Free text)

**Before being diagnosed with PE/MRI scan/visit at 1 year year postpartum did the woman**
► feel anxious (b)
► feel as if the end of the world was coming (b)
► experience severe dizziness (b)
► experience any weakness or paralysis (b)
► experience jitters/nervousness/nervous shaking (b)
► experienced any other symptoms not listed (b)
List other symptoms (FREE text)

and frozen within four to 4–6 hours. Samples are only thawed directly prior to analysis. Each new project using the biobank requires ethical approval, but no additional informed consent. At inclusion in the study, at delivery and at 1-year follow-up, maternal blood and urine are collected. At delivery, placental tissue, umbilical cord blood sample and cerebrospinal fluid are collected in addition to the above. Cerebrospinal fluid is only collected if spinal anaesthesia is administered. If the woman undergoes MRI of the brain, an extra blood sample is drawn. A detailed description of samples can be found in table 2. By merging the participants in the biobank with the GO PROVE and UP MOST database, a database and biobank is created that will also include results from organ-specific evaluation as described in table 2.

Biosamples will be analysed for prespecified biomarkers such as cerebral biomarkers, angiogenic biomarkers, cardiac biomarkers and endothelial biomarkers. In addition, biosamples will be available for future analyses of other potential biomarkers within the field of pre-eclampsia by the authors and other research groups after ethical approval and approval from the research group.

## GO PROVE and UPMOST examinations

For an overview of included examinations, see figure 1.

### MR examination of the brain, assessment of cerebral autoregulation and cognitive function scoring
#### MR brain

Investigations will be performed using 3T MR scanners at inclusion and at 1 year post partum for cases and controls. Imaging protocols contain the following sequences: 3D T1-weighted (w) MPRAGE sequence, 3D fluid-attenuated inversion recovery (FLAIR) and 2D diffusion-weighted imaging (DWI) with three directions and multiple b-values for morphological evaluation of, for example, oedema and gross pathology as well as perfusion assessment using intravoxel incoherent motion technique,[10] 2D DWI with 32 directions to evaluate white matter microstructure,[11] arterial spin labelling for perfusion assessment,[10 12] 3D time of flight MR angiography for vessel evaluation, susceptibility weighted imaging (SWI/QSM or SWAN) for assessment of haemorrhage and resting state functional MR for evaluation of neuronal networks.[13]

#### Neurocognitive assessment

To determine whether the participating women experience cognitive impairment, several instruments that measure, both subjective and objective cognition, provide a fine-grained picture of the participants' cognitive performance. Subjective cognitive function is assessed by the Cognitive Failure Questionnaire (CFQ) measuring self-reported cognitive failures in everyday life.[14] The objective cognitive test battery consists of 11 tests covering a broad range of cognitive abilities. Verbal ability is assessed with a Swedish four-alternative multiple-choice synonym test (SRB 1) and serves as an estimate of premorbid cognitive ability.[15] Mental speed is measured by Digit Symbol Coding from Wechsler Adult Intelligence Scale (WAIS IV).[16] Working memory is assessed by the letter-number sequencing task and the three-digit span tasks forward, backward and sequencing from WAIS-IV.[16] Executive functions are measured by three tasks from the Delis–Kaplan Executive Function System (D-KEFS): (1) mental shifting/flexibility with the Trail Making Test; (2) inhibition with the Stroop test; and (3) verbal fluency with Letter and Category fluency test.[17] Several aspects of episodic memory performance are measured with free and delayed recall of nouns using the Buschke Selective Reminding procedure and prospective episodic memory with the Reminding task.[18 19] All cognitive tests have good psychometric properties. The tests are administered by

**Table 1** Biochemical analyses specific for the study and in clinical routine when available

| Prenatal screening | At inclusion | At delivery and summary of analyses at discharge* |
|---|---|---|
| **If taken in clinical routine** Human chorionic gonadotropin) (c) <br> Oestriol (c) <br> Alpha-fetoprotein (c) <br> Inhibin A (c) <br> Pregnancy-associated plasma protein A (c) <br> Cell-free fetal DNA (c) | **If taken in clinical routine** <br> Haemoglobin (c) <br> Haematocrit (c) <br> WCC (white cell count) (c) <br> Platelets (c) <br> Creatinine (c) <br> AST (aspartate transaminase) (c) <br> ALT (alanine transaminase) (c) <br> Total bilirubin (c) <br> Albumin (c) <br> LDH (lactate dehydrogenase) (c) <br> Haptoglobin (c) <br> APTT (activated partial thromboplastin time) (c) <br> PR (prothrombin ratio) (c) <br> INR (international normalise ratio) (c) <br> CRP (C reactive protein) (c) <br> Oxygen saturation (d) <br> Total leucocyte count (c) <br> Urinary prot/creat ratio (c) <br> Fibrinogen (c) <br> Antithrombin (c) <br> **Specific for the study** <br> ROTEM (b) <br> Multiplate (b) <br> NT-pro BNP (c) <br> Troponin (c) | **At delivery specific for the study** <br> ROTEM (b) <br> Multiplate (b) <br> Platelets (c) <br> Troponin (c) <br> NT-pro BNP (c) <br> **Summary of analyses at discharge if taken in clinical routine** <br> Haemoglobin (g/L) (c) <br> Platelet count ($\times 10^9$/L) (c) <br> Urinary protein/creatinine ratio (mg/g) (c) <br> Creatinine (mmol/L) (c) <br> AST (IU/L) (c) <br> ALT (IU/L) (c) <br> GGT (IU/L) (c) <br> Total bilirubin (µmol/L) (c) <br> Albumin (g/L) (c) <br> LDH (IU/L) (c) <br> Haptoglobin (mg/dL) (c) <br> APTT (s) (c) <br> INR (c) <br> D-dimer (ug/L) (c) <br> CRP (mg/L) (c) |

*Lowest antepartum in hospital/highest antepartum in hospital/lowest value post partum/highest value post partum.

study personnel, at inclusion at the hospital and again at 1 year post partum.

## Transcranial cerebral Doppler

Measurement of the blood flow velocity in the middle cerebral artery is performed with transcranial Doppler technique and combined with continuous non-invasive measurement of the blood pressure using finger arterial volume clamping (calibrated with brachial blood pressure measurement) and carbon dioxide concentrations in exhaled air. The signal is obtained through the temporal window and systolic, mean and diastolic blood flow velocity and depth of measurement are registered. The measurements are performed by trained investigators, and intraobserver and interobserver variation is measured and kept below 15%. Cerebral autoregulation is determined from the response of the cerebral blood flow velocity to spontaneous fluctuations in mean blood pressure as described previously and reported as Autoregulation Index.[20] The measurements will be compared between cases and normotensive controls and between inclusion and at 1 year post partum.

## Echocardiography and endothelial dysfunction measurement
### Echocardiography

Women with pre-eclampsia and pregnant controls are examined with transthoracic echocardiography (TTE) for evaluation of systolic and diastolic function. Systolic dysfunction is defined as having an ejection fraction less than 50% or having regional hypokinesia. Diastolic dysfunction is defined as fulfilling at least 50% of the following; an average peak velocity of early diastolic transmitral flow/peak velocity of early diastolic mitral annular motion as determined by pulsed wave Doppler (E/e′)>14, septal e′ velocity <7 cm/s

**Table 2** Biological sample collection

| At inclusion | At delivery | At MRI brain | At 1 year post partum |
|---|---|---|---|
| Serum <br> EDTA plasma <br> Li/heparin plasma <br> EDTA whole blood <br> Urine | Serum <br> EDTA plasma <br> Li/heparin plasma <br> EDTA plasma umbilical cord <br> Cerebrospinal fluid <br> Placental villous tissue frozen <br> Placental villous tissue RNA later <br> Basal plate, full thickness frozen <br> Basal plate, full thickness RNA later | Serum <br> EDTA plasma | Serum <br> EDTA plasma |

or lateral e′ velocity <10 cm/s, tricuspid regurgitation velocity >2.8 m/s or left atrial volume index >34 mL/m². [21] Right ventricular function is assessed by eye-balling, tricuspid annular plane systolic excursion and right ventricular peak velocity of systolic mitral annular motion as determined by pulsed wave Doppler (RV S').

The examination is performed in proximity of inclusion and 1 year post partum.

### Peripheral artery resistance measurements

The status of the peripheral vessels can be measured with a variety of methods. In common, they are based on the interpretation of pulse pressure curve in the vessels. In this study, we use the reactive hyperemia-peripheral arterial tonometry (RH-PAT) method. The RH-PAT induces ischaemia in one arm during 5 min. The pulse pressure wave in the peripheral artery, that is, the finger, is measured before, during and after ischaemia in both arms. The difference in reaction is recorded and reflects the vascular status peripherally. [22]

The Systemic Vascular Resistance Index (SVRI) is calculated from the cardiac output and estimation of central venous pressure measured through TTE indexed to body mass. SVRI reflects the afterload, that is, the workload that the heart has to handle at a normovolemic state which in turn will be affected by the patient's blood pressure and peripheral vasoconstriction. [23]

The measurements of the heart function and the derived systemic vascular resistance retrieved from the TTE will then be compared with the measurement of peripheral vascular function as measured by the RH-PAT method and serum biomarkers clinically used to measure heart failure. The examination is performed at inclusion and again at 1 year post partum.

### Haemostasis

Haemostatic status can be assessed by laboratory analyses and by point-of-care devices as thromboelastometry (ROTEM) and impedance aggregometry (Multiplate). ROTEM is an established viscoelastic method for haemostasis testing in whole blood and a modification of traditional thromboelastography (TEG). ROTEM investigates the interaction of coagulation factors, their inhibitors, anticoagulant drugs, blood cells (specifically platelets) during clotting and subsequent fibrinolysis. [24] Multiplate measures platelet function through impedance aggregometry. [25] ROTEM and Multiplate are assessed at inclusion and again at delivery (GO-PROVE).

### Women's and partner's experiences of pre-eclampsia during pregnancy, childbirth and postnatal care
#### Questionnaires

Effects of pre-eclampsia on women's self-reported health are measured with validated instruments sent to them by email, 8–12 weeks post partum. Self-efficacy, the belief that one is capable of accomplishing a behaviour or developing a competency to succeed in reaching a specific or more general goal, is measured by the General Self-Efficacy Scale (GES). [26] Anxiety and depression are measured by

Hospital Anxiety and Depression Scale (HADS) [27] and Edinburgh Postnatal Depression Scale (EPDS). [28] Health-related quality of life is determined by the EQ-Visual Analogue Scale (EQ-VAS). [29] Childbirth experiences is measured by the Childbirth Experience Questionnaire (CEQ) [30] and levels of breastfeeding self-efficacy is assessed through the Breastfeeding Self-Efficacy Scale short form (BES). [31] Post-traumatic stress symptoms is evaluated by the post-traumatic stress disorder checklist, Diagnostic and Statistical Manual of Mental Disorders (DSM-5), Post-traumatic Stress Disorder Checklist(PCL-5). [32]

#### Interviews

Experiences of pre-eclampsia during pregnancy, childbirth and postnatal care are described through qualitative interviews. Informants will be 15–20 women who have experienced pre-eclampsia with severe features during pregnancy and their partners. [33] Informants are selected to ensure a broad range of views and experiences of the phenomenon pre-eclampsia. Face-to-face interviews [34 35] are conducted 2–3 months after delivery. An open-ended question is used, 'Please tell me of your experience of pre-eclampsia' or 'please tell me of your experience as a partner to a woman with pre-eclampsia'.

### Data processing, analysis and statistics

Results from cerebral Doppler, echocardiography and RH-PAT estimations will be calculated by two independent interpreters blinded to groups and entered manually into the database. MR data will be analysed in collaboration by neuroradiologists, blinded to groups. Blood samples will be thawed for analyses of cardiac, renal, neurological and endothelial biomarkers, analysed through standardised platforms when available and for the remaining through manual ELISA analyses in duplicate. Inter-assay and intra-assay coefficients of variation will be aimed at below 10%.

Demographics will be presented as medians or means as appropriate by distribution and by numbers and percentages as appropriate. Comparison between groups will be analysed by Student's t-test or Mann-Whitney U-test with means or medians and CIs or IQR, as appropriate according to distribution of the variables. Proportions will be analysed through $\chi^2$ test. Correlations will be analysed by Pearson's r or Spearman's rho as appropriate by distribution of the variable. Regression analyses, unadjusted and adjusted, will be performed to adjust for known confounding variables. Ten cases per variable at the lowest will be considered appropriate to avoid overfitting of the model. All statistical analyses will be performed in SPSS or R. Analyses of cognitive function will be performed through analysis of variance and multivariate analysis of variance.

Data analysis of the interview data will be conducted by phenomenology with a lifeworld approach [36] or with qualitative content analysis. [37] NVivo V.10 software (qsrinternational.com/) is used for grouping and organising the text.

## Power calculations

Uppsala and Sahlgrenska University hospitals have 4200 and 10 500 deliveries yearly respectively and Södra Älvsborgs Hospital 3100 deliveries yearly. Pre-eclampsia affects 3% of the population resulting in approximately 530 cases yearly at the centres and approximately 410 cases after exclusions according to the study protocol. For logistical reasons, all women cannot be included. Of eligible, we estimate that we will recruit around 20% for MR, transcranial Doppler and cognition examinations (70/year) at Uppsala and Sahlgrenska University hospitals and about 40/year at Sahlgrenska University hospital for cardiac and endothelial function tests. Therefore, we estimate that inclusion will take 2 years. Inclusion during 2 years at two centres will thus give around 140 MR investigations, 100 transcranial Doppler and cognition examinations and 40 TTE. We will include 60 controls for each investigation. For blood-based tests, we will use the whole cohort. We want to include more cases than controls since we want to be able to subtype cases based on pre-eclampsia phenotypes. Some specific power calculations:

### Cerebral blood flow regulation and tissue perfusion

In order to demonstrate a change of 3 mL/min×100 g in perfusion in the cortical subvolume of the brain,[38] 44 cases of pre-eclampsia and 44 normotensive controls with healthy pregnancies are required. In order to detect a difference in dynamic cerebral autoregulation index of 1.2,[20] a sample size of 32 women in each group is estimated to be required. In addition, differences in cerebral perfusion and dynamic cerebral autoregulation have been detected in the above sample sizes of 44 and 20 pre-eclampsia cases, respectively, with an equal number of normotensive controls.

### Cognition

In a meta-analysis, a difference in the objective test 'letter number sequencing task' was found when pooling results from 101 women with previous pre-eclampsia and 98 women with previous healthy pregnancies.[39]

### Patient and public involvement

To ensure that the research is relevant, acceptable and feasible from the women's and partners' point of view, patient representatives were invited to participate in the research process.[40] Focusing on women's and their partners' experiences of pre-eclampsia diagnosis and care at the same time will ensure that findings from this project will improve not only clinical outcome measures but also women's and their partners' experience.

### Ethics and dissemination

This is a cohort study with a control group with no intervention for the participants. Incidental pathological findings on examinations performed are and will be handled in accordance with clinical routine. Data are stored in a secure online database. Biobank samples are identified through the women's personal identification number and pseudonymised after identification in the biobank before analysis. Results from the study will be published in international peer-reviewed journals.

## DISCUSSION

There is a need for more individualised treatment of women with pre-eclampsia with respect to different organ manifestations of the disease. This in order to individualise treatment and care, such as an informed decision to deliver and to avoid short-term and long-term complications for the mother and infant. Currently, there are no reliable methods in clinical practice to evaluate organ dysfunction such as cardiovascular and cerebral effects of pre-eclampsia, although cerebral oedema, intracranial bleeding and pulmonary oedema, potentially secondary to heart failure are the most common causes of maternal mortality due to pre-eclampsia. There are emerging biomarkers and prediction models but none of them are used in routine care yet and they do not seem to be able to differentiate between different organ dysfunctions.[41 42]

The interdisciplinary team of obstetricians, midwives, anaesthesiologists, neuroradiologists and cardiologists enables us to link organ specialists in the field of pre-eclampsia research. This collaboration will enable important discoveries that have great potential to improve the care of these women. In addition, by employing a database created to harmonise data collection in pre-eclampsia research, biological samples and data can be shared with other research groups, enabling larger datasets for rare outcomes.[7] Prospective recruitment may imply selection bias. In this study, inclusions will mainly occur daytime by research personnel even though medical staff have the opportunity to include women after office hours (between 16:00 and 08:00). This might introduce selection bias regarding severity of disease where women with sudden onset of organ complications are more often seen and delivered after office hours. In addition, there might be a larger proportion of women with early onset pre-eclampsia recruited since women with late onset pre-eclampsia are induced shortly after diagnosis. However, national health records are available to compare demographics of the study population to the general population with pre-eclampsia.

Regarding cerebral dysfunction in pre-eclampsia, we will be able to further characterise imaging abnormalities, circulating cerebral biomarkers and cognitive function among women with pre-eclampsia and neurological impairment. Some women with pre-eclampsia show evidence of cerebral oedema in the acute phase[43 44] and long-term follow-up of women with previous pre-eclampsia demonstrate an increased number of white matter lesions.[45] There is a paucity in data regarding the connection between acute lesions and long-term effects on the brain and their implications in the development of stroke, epilepsy and dementia. Cerebral biomarkers have been reported to be increased in pre-eclampsia but the connection to objective findings on MR or cognitive function is not clear.[46 47] Hopefully, data from this study can

enable an objective assessment of neurological function in pre-eclampsia as a tool for targeting women at risk for adverse cerebral events. In addition, the long-term cerebral effects of pre-eclampsia are still an area that needs more research. There are reports of an increased risk of dementia, stroke and epilepsy after a pregnancy complicated by pre-eclampsia but the causality and importance of pre-eclampsia as an insult to the brain is not fully elucidated.[48–50] Cognitive impairment years after a pregnancy complicated by pre-eclampsia has been reported but it is not known whether this cognitive impairment exists already at diagnosis and how it changes over the first year post partum after pre-eclampsia.[51 52] Findings from this study might increase the understanding of the underlying pathophysiological mechanisms of acute cerebral injury and long-term effects.

Regarding haemostasis, our results will hopefully provide guidance as to when women with pre-eclampsia can receive spinal anaesthesia and further insight into the coagulation profile among women with severe pre-eclampsia.

For pre-eclampsia and cardiovascular function, an initial assessment of cardiovascular function with TTE and peripheral resistance measurement for assessment of endothelial dysfunction for all women in the study will aid in the risk-classification of women stratified on cardiovascular involvement.

Increased knowledge about how women as well as partners experience pre-eclampsia and the consequences could improve personalised care and follow-up after pre-eclampsia.

Pre-eclampsia is a disease only manifested in humans, hence there are no animal models of pre-eclampsia with the ability to exactly mimic the true pre-eclampsia state.[53] Therefore, it is imperative that critical laboratory and clinical observations are made on humans and human tissues. By creating a database and biobank, we will be able to make important laboratory discoveries that can be translated to clinical trials that may improve the care of women.

**Author affiliations**
[1]Institute of Health and Care Sciences, Sahlgrenska Academy, University of Gothenburg, Goteborg, Sweden
[2]Department of Anaesthesiology and Intensive Care, Institute of clinical sciences, Sahlgrenska Academy, University of Gothenburg, Gothenburg, Sweden
[3]Department of Obstetrics and Gynecology, Institute of clinical sciences, Sahlgrenska Academy, University of Gothenburg, Goteborg, Sweden
[4]Department of Obstetrics and Gynecology, Region Västra Götaland, Sahlgrenska University Hospital, Goteborg, Sweden
[5]Department of Women's and Children's Health, Uppsala University, Uppsala, Sweden
[6]Department of Obstetrics and Gynecology, University Medical Center Groningen, Groningen, The Netherlands
[7]Department of Surgical Sciences, Neuroradiology, Uppsala University, Uppsala, Sweden
[8]Department of Radiology, Institute of Clinical Sciences, Sahlgrenska Academy, University of Gothenburg and Sahlgrenska University Hospital, Gothenburg, Sweden
[9]Department of Social and Psychological Studies, Karlstad University, Karlstad, Sweden
[10]Engineering Psychology, Luleå University of Technology, Luleå, Sweden
[11]Department of Obstetrics and Gynecology, Stellenbosch University, Cape Town, South Africa

**Acknowledgements** We would like to thank Biobank Väst and Uppsala Biobank (a part of SIB) for supporting with collection of biological specimens.

**Contributors** LB and LT wrote the article. LB contributed with information about biosamples in GO PROVE, transcranial Doppler examination, MRI examinations, structure of the online database and inclusion of women in GO PROVE. LT contributed with information about questionnaires, cognitive tests, handling of biosamples and structure of the online database. MA, JO and S-ET provided the methods and contributed to the sections regarding cardiac function and endothelial dysfunction. OK provided the methods and contributed to the sections regarding coagulation. VS contributed to the section about recruitment of the population. TS and SL contributed to the Methods section about placental sampling. HE and KL provided the methods and contributed to the sections regarding women's experience of childbirth and questionnaires. KJ and TvV contributed to the sections about transcranial Doppler and cerebral autoregulation. IE contributed to the section about recruitment of participants in Uppsala (UPMOST). JW and IB-B provided the methods section and contributed to the sections about MRI. ASN provided the methods section and contributed to the sections about cognitive function tests. A-KW contributed to the section about recruitment in Uppsala and handling of biosamples. All authors revised and approved the final version of the manuscript.

**Funding** This study is supported by the Swedish Brain Fund, Märta Lundqvist's foundation, Jane and Dan Olsson's foundation, the Foundation of the Health and Medical care committee of the Region of Västra Götaland, Sweden (VGFOUREG851391), (VGFOUREG930918), Gothenburg Society of Medicine (GLS934555 and GLS878481), Sahlgrenska University Hospital, ALF Västra Götalands Region (ALFGBG-925851), Swedish Research Council (2020-01640), Swedish Society for Medical Research and Sahlgrenska University Hospital SU (2018-03591).

**Competing interests** None declared.

**Patient consent for publication** Not applicable.

**Provenance and peer review** Not commissioned; externally peer reviewed.

**ORCID iDs**
Ove Karlsson http://orcid.org/0000-0003-2020-4695
Verena Sengpiel http://orcid.org/0000-0002-3608-7430
Teresia Svanvik http://orcid.org/0000-0003-1947-679X
Helen Elden http://orcid.org/0000-0003-0000-0476
Karolina Linden http://orcid.org/0000-0002-2792-3142
Katja Junus http://orcid.org/0000-0003-4088-400X
Susanne Lager http://orcid.org/0000-0003-3556-065X
Johan Wikström http://orcid.org/0000-0002-9481-6857
Lina Bergman http://orcid.org/0000-0001-5202-9428

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
