## [Reviewer comments · BMJ Open]

ARTICLE DETAILS

TITLE (PROVISIONAL)	Study protocol; Establishment of a multi-center preeclampsia database and biobank in Sweden – GO PROVE and UP MOST, a prospective cohort study
AUTHORS	Thorgeirsdottir, Lilja; Andersson, Malin; Karlsson, Ove; Thörn, Sven-Egron; Oras, Jonatan; Sengpiel, Verena; Svanvik, Teresia; Elden, Helen; Linden, Karolina; Junus, Katja; Lager, Susanne; Enskär, Ida; van Veen, Teelkien; Wikström, Johan; Björkman-Burtscher, Isabella; Stigsdotter Neely, Anna; Wikström, Anna-Karin; Bergman, Lina

VERSION 1 – REVIEW

REVIEWER	Zhang, Yu Shanghai Jiao Tong University School of Medicine Affiliated Renji Hospital, Obstetrics & Gynecology
REVIEW RETURNED	04-Jul-2021

GENERAL COMMENTS	The aim of this ongoing clinical research is to find objective assessment of organ dysfunction such as cardiovascular and cerebral effects of preeclampsia, which are the most common causes of maternal mortality due to preeclampsia. It is very important for clinical practice. The majority of the study design are perfect, except the following items need to be clarified: 1, In the exclusion criteria, the authors did not exclude autoimmune diseases like SLE, these diseases are also the high risk pre-existing comorbidities for preeclampsia. They may also have organ dysfunction.2, For normotensive controls, the authors should give more details about how to match with the preeclampsia patients.
--

REVIEWER	Lindquist, Anthea University of Melbourne My own research unit in Melbourne has collaborated with the lead author Lina Bergman on various projects, however I have not, myself, been involved directly in any current collaborations not am involved in the study described.
REVIEW RETURNED	07-Aug-2021

GENERAL COMMENTS	This study will see the creation of an incredibly rich database with an unprecedented breadth of imaging and biological samples. Mention is made of biomarkers but there is no mention of the specific biomarkers to be explored and/or whether this will be decided and altered over time depending on emerging evidence and/or collaboration with other teams working in this field? This needs to be clarified in the manuscript. Is the intention that this database is a shared resource? Or will it only be accessible by the index research
---

	team? The protocol doesn't clarify whether there is any intention of correlating severity of disease with biological and imaging markers? The aims for sample collection are clear but it would be helpful to clarify the research question/s.
--	--

REVIEWER	Kay, Vanessa McMaster University
REVIEW RETURNED	07-Aug-2021

GENERAL COMMENTS	The manuscript is a study protocol for a multi-centre prospective cohort study in Sweden with one year follow-up including qualitative and quantitative outcomes of preeclampsia and the creation of a biobank for future research. Data collection began in 2019 and is anticipated to complete in 2023. The proposed outcomes are ambitious and include using such modalities as MRI imaging, cerebral dopplers, EEGs, neurocognitive testing, echocardiograms, and biomarkers of cerebral, cardiovascular and endothelial function. The manuscript is well-written and the outcomes have been selected with care. More information on the study methodology is needed within the manuscript for clarity. Major Comments:  - The exclusion criteria exclude women with pre-existing hypertension, cardiovascular disease or pre-existing/gestational diabetes. Excluding these women may decrease the generalizability of the results. Please add the reasoning behind excluding these women to the paper as justification. - Only a proportion of women with preeclampsia will be recruited at each site due to logistical reasons. How will this proportion be selected? Will random sampling be used? Will there be any measures in place to prevent selection bias and ensure that the included participants are similar in characteristics to the excluded participants? - The protocol should include the specific criteria used for diagnosis of preeclampsia (for example if the ACOG definition will be used) and how this was assessed (ie taken from patient chart after clinical diagnosis or separately evaluated by study team?). Will only singleton pregnancies be included or will multiple gestations also be eligible? - Specific inclusion/exclusion criteria for normotensive controls should be added to the paper as well. Are women with a history of preeclampsia in previous pregnancies but normotensive in the current pregnancy excluded? Will any matching on gestational age or maternal age be in place? Additionally, how will these women be recruited? What methods are in place to prevent selection bias? - What will be the approach to women who present with multiple pregnancies during the study recruitment period? Are they only eligible for inclusion once? Minor Comments:  - The paper states that CSF is collected at delivery. Please clarify if this will be done in the context of epidural/spinal placement or if a lumbar puncture is being done specifically for collection purposes. - Was inclusion of neonatal/offspring outcomes considered at one year? - It is clearly stated that the data analysis stage will be blinded. Will the study personnel administering the neurocognitive testing and other investigations also be blinded to preeclampsia/normotensive status?
---

VERSION 1 – AUTHOR RESPONSE

#Reviewer: 1

The aim of this ongoing clinical research is to find objective assessment of organ dysfunction such as cardiovascular and cerebral effects of preeclampsia, which are the most common causes of maternal mortality due to preeclampsia. It is very important for clinical practice. The majority of the study design are perfect, except the following items need to be clarified:

-In the exclusion criteria, the authors did not exclude autoimmune diseases like SLE, these diseases are also the high risk pre-existing comorbidities for preeclampsia. They may also have organ dysfunction.

We thank the reviewer for the positive feed-back and important comment. Regarding exclusion criteria, we have chosen to not exclude women that might be of interest regarding underlying pathophysiological pathways that do not disturb the picture of organ complications such as obesity or SLE. Thus, we have chosen not to state SLE alone as exclusion criteria but if these women in addition have renal disease, cardiac disease or cerebral disorders they are not eligible since it will be difficult to evaluate the origin of a potential pathological findings on organ evaluation. For some of our outcomes or biomarker concentrations, sub-studies in the biobank might have additional exclusion criteria that will be stated in a flow-chart for that specific article.

-For normotensive controls, the authors should give more details about how to match with the preeclampsia patients.

Since the biobank contains several sub-studies with various physiological examinations such as echocardiography and MRI brain investigations, women will be matched on a group level in these different sub-groups. In the remaining inclusions for the biobank, there will be no specific matching. We have now added this to the methods section that now reads:

“For the database and biobank as a whole there will be no matching. For specific sub-studies there might be matching on a group level for gestational age at examination.”

Page 9, 174-175

#Reviewer: 2

This study will see the creation of an incredibly rich database with an unprecedented breadth of imaging and biological samples.

-Mention is made of biomarkers but there is no mention of the specific biomarkers to be explored and/or whether this will be decided and altered over time depending on emerging evidence and/or collaboration with other teams working in this field? This needs to be clarified in the manuscript.

We thank the reviewer for the kind words and for this important comment. We will analyze both pre-defined biomarkers and also collect bio-samples for future biomarkers that might be of interest including collaboration with other researchers. The section now reads;

“Bio-samples will be analyzed for pre-specified biomarkers such as cerebral biomarkers, angiogenic biomarkers, cardiac biomarkers and endothelial biomarkers. In addition, bio-samples will be available for future analyses of other potential biomarkers within the field of preeclampsia by the authors and other research groups after ethical approval and approval from the research group.”

Page 10, lines 202-206

-Is the intention that this database is a shared resource? Or will it only be accessible by the index research team?

Thank you for raising this important point. We welcome collaborations which is also in line with the

standardized form of collecting bio-samples and clinical information. We refer to the discussion above that is now also included in the section on page 10, lines 202-206.

-The protocol doesn't clarify whether there is any intention of correlating severity of disease with biological and imaging markers? The aims for sample collection are clear but it would be helpful to clarify the research question/s.

Thank you for these remarks. Within the various investigations and analyses of biomarkers, there will be potential to explore correlation of disease phenotype with organ impairment shown on ultrasound, MRI, blood-based analyses or other modalities to explore potential diagnostic tools and/or investigate pathophysiological mechanisms.

We decided to include our overall aims rather than specific research questions in the protocol since there would be much detail with all planned research questions. In addition, since this is an ongoing biobank and database, additional research questions will hopefully arise within the research group or in collaboration with other researchers and the biobank and database will be a future source to explore various research questions. If the reviewer wish, we could add a supplementary section with examples of specified research questions.

#Reviewer: 3

The manuscript is a study protocol for a multi-centre prospective cohort study in Sweden with one year follow-up including qualitative and quantitative outcomes of preeclampsia and the creation of a biobank for future research. Data collection began in 2019 and is anticipated to complete in 2023. The proposed outcomes are ambitious and include using such modalities as MRI imaging, cerebral dopplers, EEGs, neurocognitive testing, echocardiograms, and biomarkers of cerebral, cardiovascular and endothelial function. The manuscript is well-written and the outcomes have been selected with care. More information on the study methodology is needed within the manuscript for clarity.

Major Comments:

-The exclusion criteria exclude women with pre-existing hypertension, cardiovascular disease or pre-existing/gestational diabetes. Excluding these women may decrease the generalizability of the results. Please add the reasoning behind excluding these women to the paper as justification.

We thank the reviewer for the positive feedback. The reason for the above exclusion criteria is due to the confounding these underlying chronic diseases would confer, please see comment to reviewer 1. Results from examinations such as MRI or ultrasound would be difficult to interpret if the woman already had underlying chronic disease with affection of that specific organ.

-Only a proportion of women with preeclampsia will be recruited at each site due to logistical reasons. How will this proportion be selected? Will random sampling be used? Will there be any measures in place to prevent selection bias and ensure that the included participants are similar in characteristics to the excluded participants?

We thank the reviewer for this excellent point. The reason why we approach a proportion of the women getting the diagnosis is largely due to the fact that our study personnel work office hour. We do get aid from personnel working at the hospital, but we know that women are not approached at the same rate during after-hours as they are during office hours. There is a risk that this will introduce selection bias such as the fact that women that are more severely affected will be delivered sooner and often at night. But also, women with early onset preeclampsia might be over-represented in the database since they are often pregnant longer and not induced soon after diagnosis. Though, inclusion of women on an individual basis is at random according to above strategy. To characterize our population, we will be able to compare them to the general pregnant population registered in our national quality register the Swedish Pregnancy register to evaluate generalizability. We have added this aspect to limitations in the discussion section that now reads;

“Prospective recruitment may imply selection bias. In this study, inclusions will mainly occur daytime by research personnel even though medical staff have the opportunity to include women after office hours (between 4 p.m and 8 a.m.). This might introduce selection bias regarding severity of disease where women with sudden onset of organ complications are more often seen and delivered after office hours. In addition, there might be a larger proportion of women with early onset preeclampsia recruited since women with late onset preeclampsia are induced shortly after diagnosis. However, national health records are available to compare demographics of the study population to the general population with preeclampsia.

Page 18, lines 389-397

-The protocol should include the specific criteria used for diagnosis of preeclampsia (for example if the ACOG definition will be used) and how this was assessed (ie taken from patient chart after clinical diagnosis or separately evaluated by study team?).

Thank you for this comment. Regarding the diagnosis of preeclampsia this is established according to Swedish guidelines for hypertensive disorders in pregnancy but in addition, we require significant proteinuria for research purposes. We have now added this as a clarification and it reads;

“The diagnosis of preeclampsia will be established according to Swedish guidelines for hypertensive disorders in pregnancy (www.sfog.se) that generally follows international guidelines such as ISSHP, NICE and ACOG with the addition that we will require proteinuria to be present to limit the heterogeneity of the population.”

Page 9, lines 176-179

-Will only singleton pregnancies be included or will multiple gestations also be eligible?

Multifetal pregnancies are not an absolute contraindication. Though, for the present research questions, we only include women with singleton pregnancies. Should a specific research question for multifetal pregnancies arise we have the possibility also to include these women.

-Specific inclusion/exclusion criteria for normotensive controls should be added to the paper as well. Are women with a history of preeclampsia in previous pregnancies but normotensive in the current pregnancy excluded? Will any matching on gestational age or maternal age be in place? Additionally, how will these women be recruited? What methods are in place to prevent selection bias?

Thank you for noticing this. We have chosen not to state this as an exclusion criterion for the biobank and database since there might be future research questions regarding women with previous pregnancies complicated by preeclampsia that are now normotensive. Each sub-study might select the appropriate control group according to various background characteristics such as previous preeclampsia or other pre-existing disorders due to the detailed information provided in the database.

-What will be the approach to women who present with multiple pregnancies during the study recruitment period? Are they only eligible for inclusion once?

Thank you for this remark. Each woman is only eligible once. This has now been clarified in the methods section and now reads;

“Women who have previously been included in the study will not be eligible for their next pregnancy.”

Page 9, lines 172-173

Minor Comments:

-The paper states that CSF is collected at delivery. Please clarify if this will be done in the context of epidural/spinal placement of if a lumbar puncture is being done specifically for collection purposes.

Thank you for bringing this up. CSF is only collected when the woman undergoes a C-section and a

spinal anesthesia is administered. We've clarified this in the manuscript which now reads:

"Cerebrospinal fluid is only collected if spinal anesthesia is administered."

Page 10, line 197

-Was inclusion of neonatal/offspring outcomes considered at one year?

We have not included long-term follow up prospectively for neonatal/offspring outcomes. Though, through the extensive Swedish national and quality registers we will be able to follow our population. This will require a new ethical permission.

-It is clearly stated that the data analysis stage will be blinded. Will the study personnel administering the neurocognitive testing and other investigations also be blinded to preeclampsia/normotensive status?

You raise an important point. Regarding the neurocognitive tests, the study personnel administering them are a part of the team recruiting women to the study and can therefore not be blinded to their status. The data from the tests are recorded in an online database and the professor in neuropsychology that will interpret the test results will be blinded to groups. Regarding the MRI investigations, the investigators at the MRI machine will not be blinded to groups since women with preeclampsia and in particular preeclampsia with severe features require closer surveillance and in addition, they often have an IV access and are hospitalized. Though, the professors in neuroradiology that will interpret the data will be blinded to groups.

As long as blinding is possible, such as in interpretation of data, analyses of biological samples and interpretation of these results, investigators will be blinded to groups to reduce detection bias.

VERSION 2 – REVIEW

REVIEWER	Zhang, Yu Shanghai Jiao Tong University School of Medicine Affiliated Renji Hospital, Obstetrics & Gynecology
REVIEW RETURNED	12-Sep-2021

GENERAL COMMENTS	The aim of this ongoing clinical research is to find objective assessment of organ dysfunction such as cardiovascular and cerebral effects of preeclampsia, which are the most common causes of maternal mortality due to preeclampsia. They use standard methods to evaluate organ dysfunctions, which will provide clinical data to normalize organic functional evaluation of preeclampsia. About the study design, the following issues still need to be explained: 1, Could the authors estimate how many patients and normal pregnant women they are going to recruit according to the annual delivery number and incidence of preeclampsia in the three research centers?2, According to the protocol, the patients should do many examinations, which might be difficult to complete and even to recruit patients and healthy pregnant women. How have the researchers done to promote the protocol being followed.3, In the exclusion criteria, the authors excluded women with pre-existing hypertension, cardiovascular disease or pre-existing/gestational diabetes, but they did not exclude autoimmune diseases like SLE, these diseases are also high risk factors for organ dysfunction in preeclampsia. But if all these comorbidities are excluded, the generalizability will be limited. The authors should explain why they decide these exclusion criteria.4, For normotensive controls, the authors should give more details
--

	about how to match with the preeclampsia patients.
REVIEWER	Kay, Vanessa McMaster University
REVIEW RETURNED	01-Sep-2021
GENERAL COMMENTS	The authors have responded appropriately to the previous reviewer comments.

VERSION 2 – AUTHOR RESPONSE

#Reviewer 3

The aim of this ongoing clinical research is to find objective assessment of organ dysfunction such as cardiovascular and cerebral effects of preeclampsia, which are the most common causes of maternal mortality due to preeclampsia. They use standard methods to evaluate organ dysfunctions, which will provide clinical data to normalize organic functional evaluation of preeclampsia. About the study design, the following issues still need to be explained:

1, Could the authors estimate how many patients and normal pregnant women they are going to recruit according to the annual delivery number and incidence of preeclampsia in the three research centers?

The estimated sample size is discussed in Power calculations, page 16, lines 336-339 and reads:

“Uppsala and Sahlgrenska University hospitals have 4,200 and 10,500 deliveries yearly respectively and Södra Älvsborgs Hospital 3,100 deliveries yearly. Preeclampsia affects 3% of the population resulting in approximately 530 cases yearly at the centers and approximately 410 cases after exclusions according to the study protocol.”

2, According to the protocol, the patients should do many examinations, which might be difficult to complete and even to recruit patients and healthy pregnant women. How have the researchers done to promote the protocol being followed.

In general recruitment of subjects to research studies in our country is facilitated by rigorous national ethical approval routines and well-designed informed consent forms expressing the content of a scientific study not mainly in legal and medical terms but in language appraisable by the subjects. All subjects are addressed individually by trained research staff to facilitate inclusion and exclusion. Each step in the protocol including imaging is integrated in research routines approved by the hospitals facilitating that the protocol is being followed. Subjects can whenever and without giving a reason discontinue participation in the study. From our experience this is usually seldom the case. We are thus convinced that the sample sizes given under the section on power calculation can be reached. We give in this section the different sub-populations that will undergo special investigations including their sizes and this section can be found on page 16, lines 339-346 and reads;

“For logistical reasons, all women cannot be included. Of eligible, we estimate that we will recruit around 20% for MR, transcranial doppler and cognition examinations (70/year) at Uppsala and Sahlgrenska University hospitals and about 40/year at Sahlgrenska University hospital for cardiac and endothelial function tests. Therefore, we estimate that inclusion will take two years. Inclusion during two years at two centers will thus give around 140 MR investigations, 100 transcranial doppler and cognition examinations and 40 TTE. We will include 60 controls for each investigation. For blood based tests we will use the whole cohort.”

3, In the exclusion criteria, the authors excluded women with pre-existing hypertension,

cardiovascular disease or pre-existing/gestational diabetes, but they did not exclude autoimmune diseases like SLE, these diseases are also high risk factors for organ dysfunction in preeclampsia. But if all these comorbidities are excluded, the generalizability will be limited. The authors should explain why they decide these exclusion criteria.

Regarding exclusion criteria, we have chosen not to exclude women who might be of interest regarding underlying pathophysiological pathways that do not disturb the picture of organ complications such as obesity or SLE. Thus, we have chosen not to state SLE as exclusion criteria in itself, however, if these women in addition have renal, cardiac or cerebral disorders or complications based on such a disease, they are not eligible since it will be difficult to evaluate the origin of a potential pathological finding in organ evaluation. For some of our outcomes or biomarker concentrations, sub-studies considering biobank data might have additional exclusion criteria which will be stated in flow-charts in the specific publications.

4, For normotensive controls, the authors should give more details about how to match with the preeclampsia patients.

Details on the matching are provided in the methods section, page 9, lines 174-175 and reads;

“For the database and biobank as a whole there will be no matching. For specific sub-studies there might be matching on a group level for gestational age at examination.”

Since gestational age is problematic from the point of view that it is not a true confounder for most outcomes but rather a mediator, we have chosen to match for this on a group level. Since there will be different women participating in different sub-studies where MRI, echocardiography etc are included, these different sub-groups will have their own matching on gestational length. There will be no case-case matching. In addition, we have chosen not to match for any other background data since this also might pose methodological problems when interpreting the data. We will rather use adjusted models to explore the importance of different co-variables in the population.